# Mental Health of Japanese Workers: Amotivation Mediates Self-Compassion on Mental Health Problems

**DOI:** 10.3390/ijerph191710497

**Published:** 2022-08-23

**Authors:** Yasuhiro Kotera, Kenichi Asano, Hiromasa Kotera, Remi Ohshima, Annabel Rushforth

**Affiliations:** 1School of Health Sciences, University of Nottingham, Nottingham NG7 2RD, UK; 2Department of Psychological Counseling, Faculty of Psychology, Mejiro University, Tokyo 161-0032, Japan; 3Department Linguistics, University of Potsdam, 14469 Potsdam, Germany; 4College of Health, Psychology and Social Care, University of Derby, Derby DE22 1GB, UK

**Keywords:** Japanese workers, mental health, self-compassion, amotivation, mediation

## Abstract

Workplace mental health is a cause for concern in many countries. Globally, 78% of the workforce experienced impairment of their mental health in 2020. In Japan, more than half of employees are mentally distressed. Previously, research has identified that self-compassion (i.e., being kind and understanding towards oneself) and work motivation were important to their mental health. However, how these three components relate to each other remains to be elucidated. Accordingly, this study aimed to examine the relationship between mental health problems, self-compassion and work motivation (i.e., intrinsic motivation, extrinsic motivation and amotivation). A cross-sectional design was employed, where 165 Japanese workers completed self-report scales regarding those three components. A correlation and path analyses were conducted. Mental health problems were positively associated with amotivation and negatively associated with age and self-compassion. While intrinsic motivation and extrinsic motivation did not mediate the impact of self-compassion on mental health problems, amotivation did. The findings can help managers and organizational psychologists help identify effective approaches to improving work mental health.

## 1. Introduction

In recent years, challenging mental health has been increasingly reported in organizations worldwide. Globally, 78% of the workforce experienced an impairment of their mental health in 2020 [1]. In the United Kingdom (UK), mental health problems account for 50% of all workplace health problems, and 55% of them (who suffer from mental health problems) reported a negative impact from COVID-19 [2]. Likewise, increasing rates of mental health problems such as anxiety and depression were identified, associated with a compromised quality of life, in a Singapore national survey, where 72% (*n* = 4052) were employees [3]. About half of Chinese healthcare workers have reported high levels of depression and anxiety since the outbreak of COVID-19 [4]. Moreover, Hungarian workers reported increased stress and mental fatigue, derived from new work arrangements including working from home [5]. Literature reviews have identified that working from home can be a stressor for some employees, associated with stress, depression and fatigue [6,7]. In the occupational context, workplace mental health is considered as essential for the quality of professional life and work performance [8], and it is aligned with the United Nation’s 17 sustainable development goals targeting good health and well-being [9]. Commonly, mental health is regarded as a dynamic emotional state that involves harmonization with universal world values, associated with empathy for others [10]. While these debates on the definition of mental health demonstrate an awareness of cross-cultural differences in mental health, an empirical evaluation of mental health in different cultures remains to be conducted [10,11]. In particular, the mental health status in non-WEIRD (Western, Educated, Industrialized, Rich and Democratic) areas needs to be understood more [12].

An understanding of mental health in different cultures is valuable because clinical practice has become increasingly internationalized these days [13], and culturally unaware practices and communities exclude people from marginalized backgrounds [14,15]. To offer inclusive care to diverse people, a cross-cultural understanding of mental health is necessary. For example, higher levels of mental health shame in Irish students than in UK students were suggested to be attributable to Irish collectivism (relative to UK) [11]. A higher level of mental distress in Malaysian university students than in UK students may be explained by Malaysian hierarchical culture when compared to the UK’s flat society [16]. A higher level of self-compassion (i.e., being kind and understanding towards oneself in difficult times) in Indonesian students than in UK students was considered relevant to Indonesian quality-oriented culture as opposed to UK’s success-focused culture [17]. Cultures are one explanatory factor for the mental health of people who are associated with them.

### 1.1. Poor Mental Health in Japanese Workforce

For decades, high rates of mental health problems have been consistently reported in Japanese organizations [18]. A recent national survey of mental health among Japanese employees revealed that 54.2% of them reported being distressed by work-related stressors that derived from a high workload (42.5%), troubles in the work relationship (35.0%) and the contents of the work (30.9%) [19]. Moreover, job strain, effort/reward imbalance and poor co-worker support were rather common in Japanese workplaces and were related to reduced employee wellbeing [20]. Poor occupational mental health remains a cause for concern during the COVID-19 pandemic [21]. More specifically, their working hours are long, which is associated with poor mental health such as depression and suicidal ideation [22]. Indeed, as reported in a meta-analysis [23], long hours spent alone may not be a detrimental factor in poor mental health (e.g., presenteeism); however positive associations with poor mental health have been reported among Japanese workers [24,25]. A better understanding of work mental health bears particular importance as the workforce is decreasing, which is associated with the fast-ageing society [26].

### 1.2. Self-Compassion

Self-compassion is commonly regarded as the tendency to be understanding and kind towards oneself in difficult times and circumstances [27]. Self-compassion is associated with many positive outcomes, including better mental health and ethical judgement [28,29,30,31]. In general, self-compassionate individuals cope better with challenging situations [32] and are more open to accepting negative emotional and cognitive experiences [33]. The importance of self-compassion in relation to mental health has been reported in many different cultures including Japan [34,35,36].

A meta-analysis of self-compassion interventions identified that self-compassion was effective at reducing mental health problems [37]. Likewise, a systematic review synthesizing the evidence for the organizational application of self-compassion reported that self-compassion helped reduce workplace mental distress [29]. In Japan, a self-compassion randomized controlled trial was conducted, and the intervention group (who received self-compassion training) showed a significantly greater reduction in rumination than the control group (who received a lecture on communication) [38]. These findings suggest that cultivating self-compassion is conducive to mental health.

Previous research attempted to elucidate this association between self-compassion and improved mental health. For example, resilience was identified as a significant mediator between self-compassion and mental health among healthcare students in China [39] and professional counsellors in Malaysia [40]. Emotion regulation difficulties were found to be a mediator between self-compassion and mental health among Australian adolescents [41]. Similarly, rumination and worry mediated the self-compassion–mental health relationship among Belgian students [42]. Among Japanese workers, self-compassion’s mediation of mental health shame and mental health problems was discovered [43]; however, other relationships remained to be evaluated.

### 1.3. Work Motivation

Work motivation refers to why employees engage in work and what makes them continue to do so [44], and it is related to positive organizational outcomes such as productivity and wellbeing [45]. Work motivation is well-recognized in Japanese organizations [46]. One established motivation theory is the Self-Determination Theory (SDT), which classifies motivation into intrinsic motivation (based upon curiosity and fun), extrinsic motivation (driven by external rewards such as money and fame) or amotivation (no willingness to engage in work) [45]. Intrinsic motivation is often associated with better mental health, a lower level of mental health shame and better ethical judgement, whereas extrinsic motivation is associated with poorer mental health, a higher level of shame and poorer ethical judgement [28,47]. Amotivation is positively associated with and a predictor of mental health problems [48,49]. In general, employees who are passionate about work tend to have a high level of wellbeing, are open to acknowledging negative emotions, and maintain ethical judgements, whilst employees who are predominantly driven by money, fame or status tend to be distressed, view mental health problems negatively, and make poor ethical judgements. These findings highlight the fact that work motivation is relevant to both organizational and individual outcomes.

### 1.4. Work Motivation on Pathway from Self-Compassion to Mental Health

Additionally, self-compassion has been associated with work motivation [50,51]. Neff et al. [52] found that undergraduate students’ self-compassion was positively related to their mastery of goals, suggesting that self-compassionate individuals are better able to see failure as a learning opportunity and re-engage with tasks following failure, thus potentially reducing fear of failure and increasing perceived competency. These characteristics correspond to intrinsic motivation [28]. Self-compassion was not associated with actual competency, suggesting that it impacts these motivational patterns. Interventional studies have indicated that self-compassion training can improve motivational outcomes in students [53,54,55]. A two-week clinical intervention significantly increased self-compassion in university students, with sustained high levels of self-compassion at the six-month follow-up [53]. This increase in self-compassion was associated with an improved motivation to learn, personal growth, self-efficacy and impulse control [53]. This study indicates the potential for intrinsic motivation to be cultivated through self-compassion interventions. In a study of 120 postgraduate students, Kotera et al. saw that intrinsic and extrinsic motivations were positively associated with engagement, that amotivation was negatively associated with engagement, and that a higher self-compassion moderated and strengthened the pathway from extrinsic motivation to intrinsic motivation [47,56]. Despite this body of literature, it is still unclear how exactly self-compassion influences mental health in a workplace setting. This collection of findings suggests that work motivation may mediate the impact of self-compassion on mental health. Moreover, which type of work motivation mediates the pathway from self-compassion to mental health, and which type does not, remains to be evaluated.

### 1.5. Study Aims

This study aimed to evaluate the relationship between mental health problems, self-compassion and work motivation (i.e., intrinsic motivation, extrinsic motivation and amotivation) in Japanese employees. For mental health problems, we focused on depression, anxiety and stress because these are common symptoms in the workplace [44]. Two aims were established.

Aim 1: To identify correlational relationships between mental health problems, self-compassion and work motivation.

Aim 2: To evaluate which type of work motivation mediates self-compassion’s impact on mental health problems.

## 2. Materials and Methods

### 2.1. Participants

Japanese workers were recruited from the researchers’ network. Participants had to (i) be at least 18 years old, (ii) engage with work at an organization in Japan three days or more per week, and (iii) have a minimum work experience of six months. One hundred and sixty-five Japanese workers (125 males, 40 females; Age 47.20 ± 11.85, range 20–73 years old) completed the scales. No queries or complaints were reported from the participants. Our sample had more males (76%) than the general employee population in Japan (56%; National Statistics Center, 2015). The industries that the Japanese employees worked in were education, construction (*n* = 12, 7.3%, respectively), retail, technologies (*n* = 11, 6.7%, respectively), wholesale and transit (*n* = 8, 4.9%, respectively), and the rest included finance, manufacturing and hospitality. Their highest education qualifications included a higher education degree (*n* = 110, 66%), further education degree (*n* = 28, 17%), and high school or middle school diploma (*n* = 26, 16%). This sample was originally used in a Dutch-Japanese comparative study [57].

### 2.2. Procedure

Ethical approval was granted by the university research ethics committee. Once the consent form was submitted, participants received a link to the online self-report scales.

Prior to the analyses, data were screened for outliers and distribution. First, correlation analyses were conducted to evaluate the relationships between mental health problems (depression, anxiety and stress), self-compassion and work motivation (Aim 1). Next, path analyses were performed to identify which type of work motivation mediated self-compassion’s impact on mental health problems. Analyses were performed using SPSS 27.0 created by IBM Corp (Armonk, NY, USA) [57] and Process Macro outlined by Hayes [58].

### 2.3. Instruments

The Depression Anxiety and Stress Scale 21 (DASS-21) was used to measure mental health problems. DASS-21 consists of 21 items and is a shortened version of the DASS-42 [59]. These items are divided into three subscales targeting common mental health problems (seven items each): depression (e.g., ‘I found it difficult to work up the initiative to do things’), anxiety (e.g., ‘I was worried about situations in which I might panic and make a fool of myself’) and stress (e.g., ‘I felt that I was using a lot of nervous energy’). Responses are made on a four-point Likert scale (from ‘0′ being ‘Did not apply to me at all’ to ‘3′ being ‘Applied to me very much, or most of the time’). DASS-21 has good reliability (α = 0.87–0.94) [60].

Self-compassion was examined with the Self-Compassion Scale-Short Form (SCS-SF) [61], which is a shortened 12-item version of the 26-item Self-Compassion Scale [62]. The SCS-SF evaluates how kind and understanding the employee can be to themselves in difficult times [62]. The 12 items, such as ‘When something painful happens I try to take a balanced view of the situation’, are responded to on a five-point Likert scale (from ‘1′ being ‘Almost never’ to ‘5 being ‘Almost always’). The SCS-SF has good reliability (α ≥ 0.86) [61].

Lastly, work motivation was appraised using the Work Extrinsic and Intrinsic Motivation Scale (WEIMS). Based on the Self-Determination Theory (SDT) [63], this 18-item scale assesses three types of work motivation: extrinsic motivation, intrinsic motivation and amotivation. Six subscales are embedded in WEIMS (from the least autonomous motivation to the most): (1) amotivation (e.g., ‘I don’t know why, we are provided with unrealistic working conditions.’), (2) external regulation (e.g., ‘Because it allows me to earn money’), (3) introjected regulation (e.g., ‘Because I want to be very good at this work, otherwise I would be very disappointed.’), (4) identified regulation (e.g., ‘Because I chose this type of work to attain my career goals.’), (5) integrated regulation (e.g., ‘Because it is part of the way in which I have chosen to live my life.’) and (6) intrinsic motivation (e.g., ‘For the satisfaction I experience from taking on interesting challenges.’). Participants respond to all items on a seven-point Likert-scale (from ‘1′ being ‘Does not correspond at all’ to ‘7′ being ‘Corresponds exactly’). All of the subscales have adequate-good reliability (α = 0.64–0.83; Tremblay et al., 2009). For this study, the four extrinsic motivation subscales, (2) external regulation to (5) integrated regulation, were averaged to determine ‘extrinsic motivation’ [63,64].

## 3. Results

No outliers were identified when assessed by inspection of a boxplot. All variables demonstrated a high internal consistency (α > 0.70). Descriptive statistics are summarized in Table 1.

### 3.1. Relationships between Mental Health Problems, Self-Compassion and Work Motivation (Aim 1)

Pearson correlations were performed to examine the relationships between mental health problems, self-compassion and work motivation in Japanese workers (Table 2). For gender, point biserial correlations were calculated (0 = female, 1 = female).

Mental health problems were positively associated with amotivation, and negatively associated with age and self-compassion. Self-compassion was positively associated with intrinsic motivation and negatively associated with amotivation. Intrinsic motivation was positively associated with extrinsic motivation and negatively associated with amotivation. Lastly, extrinsic motivation was negatively associated with amotivation. Among the demographic variables (age and gender), age was positively associated with gender and negatively associated with mental health problems (older workers tended to be male and have fewer mental health problems).

### 3.2. Mediation of Motivation for Self-Compassion in Mental Health Problems (Aim 2)

To identify which type of work motivation (i.e., intrinsic motivation, extrinsic motivation and amotivation) mediated self-compassion’s impact on mental health problems, three sets of path analyses were conducted, using the Model 4 in the Process macro [58] with 5000 bootstrapping re-samples and bias-corrected 95% confidence intervals (CIs) for indirect effects.

#### 3.2.1. Intrinsic Motivation

Intrinsic motivation did not mediate the relationship between self-compassion and mental health problems: incomplete mediation (Figure 1). The pathways from self-compassion to intrinsic motivation (b = 0.74, *p* = 0.0003, BCa CI [0.34, 1.14]) and from self-compassion to mental health problems (Indirect effects b = −24.69, *p* < 0.0001, BCa CI [−31.07, −18.32]; Total effects b = −24.65, *p* < 0.0001, BCa CI [−30.76, −18.54]) were significant. However, the one from intrinsic motivation to mental health problems was not significant (b = 0.06, *p* = 0.96).

#### 3.2.2. Extrinsic Motivation

Likewise, extrinsic motivation did not mediate the relationship between self-compassion and mental health problems either: incomplete mediation (Figure 2). Only the pathway from self-compassion to mental health problems (Indirect effects b = −24.63, *p* < 0.0001, BCa CI [−30.79, −18.46]; Total effects b = −24.65, *p* < 0.0001, BCa CI [−30.76, −18.54]) was significant. The pathways from self-compassion to extrinsic motivation (b = 0.18, *p* = 0.16). and from extrinsic motivation to mental health problems (b = −0.13, *p* = 0.94) were not significant.

#### 3.2.3. Amotivation

Amotivation mediated the relationship between self-compassion and mental health problems (Figure 3). All pathways were significant: from self-compassion to mental health problems (Indirect effects b = −16.85, *p* < 0.0001, BCa CI [−23.26, −10.44]; Total effects b = −24.65, *p* < 0.0001, BCa CI [−30.76, −18.54]); from self-compassion to amotivation (b = −1.23, *p* < 0.0001, BCa CI [−1.59, −0.86]); and from amotivation to mental health problems (b = 6.36, *p* < 0.0001, BCa CI [3.93, 8.80]).

## 4. Discussion

This study aimed to evaluate the relationships between mental health problems, self-compassion and work motivation in Japanese workers. Mental health problems (depression, anxiety and stress) were positively associated with amotivation, and negatively associated with age and self-compassion (Aim 1). While intrinsic motivation and extrinsic motivation did not mediate the impact of self-compassion on mental health problems, amotivation did (Aim 2). The findings are discussed below.

As expected, self-compassion was negatively associated with mental health problems and positively associated with intrinsic motivation, supporting pre-existing research in the field [28,65,66]. Additionally, amotivation was negatively associated with self-compassion and positively associated with mental health problems [67]. This shows that Japanese workers with poor mental health tended to have higher levels of amotivation, implying that mentally distressed employees are not motivated to work. This is consistent with previous research that found an association with amotivation and emotional exhaustion [68] and depression [69].

Furthermore, the path analysis showed that amotivation mediated the relationship between self-compassion and mental health; however, extrinsic and intrinsic motivation did not. This suggests that when an employee has low self-compassion that is impacting their mental health, this impact includes an increase of amotivation, but not an impact on extrinsic motivation or intrinsic motivation. These results indicate that self-compassion helps workers to be less amotivated, despite not increasing intrinsic or extrinsic motivation, implying that self-compassion impacts work motivation when autonomy is low (i.e., amotivation), more than when autonomy is medium or high (i.e., extrinsic or intrinsic motivation). No significant mediation with extrinsic and intrinsic motivation as the mediator may be explained by the Japanese working culture, which values working hard highly [70]. Taking care of oneself is not yet accepted in many Japanese organizations; therefore, self-compassion’s impact on mental health did not include any impact from extrinsic or intrinsic motivation [71]. On the other hand, when motivation was problematically low, Japanese employees felt that they needed to take care of themselves. These findings show that the incorporation of self-compassion training can mitigate amotivation in employees in the Japanese population and improve mental health. This has clinical implications regarding interventions aimed at improving motivational issues. Traditionally motivational issues are addressed via interventions aimed at enhancing extrinsic motivation such as ‘Pay for Performance’ [72], which may not be effective for those with low self-compassion experiencing amotivation, and this may account for some of the lack of efficacy of these interventions. Self-compassion training can target this group in Japanese organizations. Poor work motivation is a cause for concern in many Japanese companies [73]. Considering the high level of shame associated with mental health problems and the busy work schedule among Japanese workers [43], a brief online training session that can be completed in a private setting may be recommended [50]. This type of new approach needs to be implemented and evaluated in Japanese workplaces.

While this study offers helpful insights into the mental health of Japanese workers, several limitations should be noted. First, in addition to the relatively small sample size, our participants were limited to those who felt comfortable responding to an online survey. More inclusive ways of recruiting need to be implemented in future research. Second, as discussed, the use of self-reporting scales may be susceptible to response biases. Third, our study employed a cross-sectional design, and the causality of these associations was therefore not evaluated. Fourth, the accurate representation of some of the scales used in this study remains to be further evaluated (e.g., SCS-SF [74]).

## 5. Conclusions

Our findings highlighted the relevance of amotivation to self-compassion and mental health problems among Japanese workers. The positive impact of self-compassion training may be maximized when the training is used for workers who have a high level of amotivation. As poor work motivation is a critical issue in many organizations, our findings can inform a more effective way to implement self-compassion training for Japanese workers.

## Figures and Tables

**Figure 1 ijerph-19-10497-f001:**
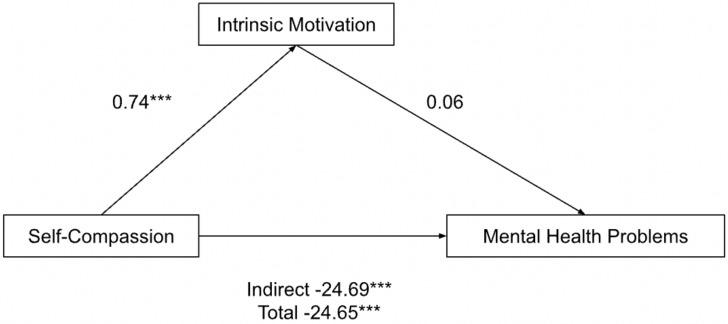
Parallel mediation model: self-compassion as a predictor of mental health problems (depression, anxiety and stress), mediated by intrinsic motivation (incomplete). *** *p* < 0.001.

**Figure 2 ijerph-19-10497-f002:**
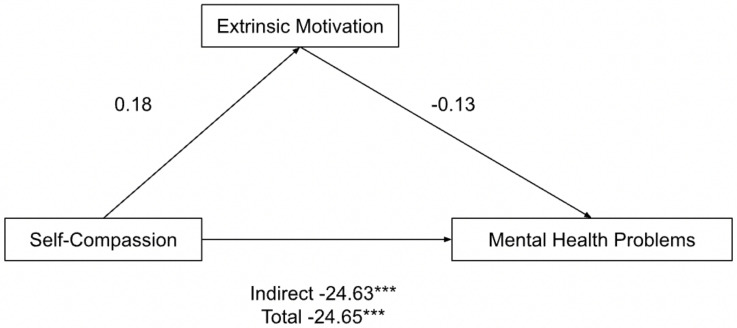
Parallel mediation model: self-compassion as a predictor of mental health problems (depression, anxiety and stress), mediated by extrinsic motivation (incomplete). *** *p* < 0.001.

**Figure 3 ijerph-19-10497-f003:**
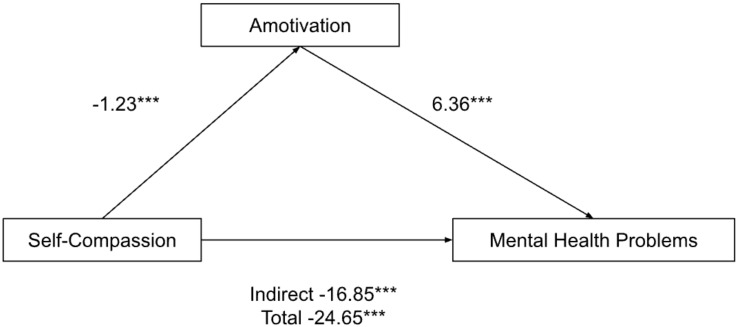
Parallel mediation model: self-compassion as a predictor of mental health problems (depression, anxiety and stress), mediated by amotivation. *** *p* < 0.001.

**Table 1 ijerph-19-10497-t001:** Descriptive statistics.

	Japanese Workers (*n* = 165)		
	M	SD	α
Mental Health Problems	22.29	22.93	0.96
Self-Compassion	3.02	0.49	0.77
Intrinsic Motivation	3.75	1.32	0.84
Extrinsic Motivation	4.10	0.82	0.80
Amotivation	2.85	1.30	0.82

Mental Health Problems = Depression, Anxiety and Stress.

**Table 2 ijerph-19-10497-t002:** Correlations between mental health problems, mental health shame, self-compassion and motivation in Japanese workers.

		1	2	3	4	5	6	7
1	Age	-						
2	Gender (0 = F, 1 = M)	0.34 **	-					
3	Mental Health Problems	−0.18 *	−0.03	-				
4	Self-Compassion	0.09	0.01	−0.53 **	-			
5	Intrinsic Motivation	−0.02	−0.01	−0.14	0.28 **	-		
6	Extrinsic Motivation	−0.04	−0.10	−0.06	0.11	0.67 **	-	
7	Amotivation	0.01	0.15	0.53 **	−0.47 **	−0.33 **	−0.18 *	-

* *p* < 0.05, ** *p* < 0.01. Mental Health Problems = Depression, Anxiety and Stress.

## Data Availability

The data that support the findings of this study are available on request from the corresponding author. The data are not publicly available due to privacy or ethical restrictions.

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
