# Peer review of "Mental Health of Japanese Workers: Amotivation Mediates Self-Compassion on Mental Health Problems"

_ijerph, 2022, doi:10.3390/ijerph191710497_

Round 1

Reviewer 1 Report

In this study Authors have highlighted some important observations regarding the relationship between mental health problems, self-compassion and work motivation. Authors have conducted this study involving 165 participants from japan and described in detail the relevance of amotivation with self-compassion and mental health problems among Japanese workers.. In order to recommend this article for editorial decision, Authors need to clarify some minor points:

1) In line 144 Authors have stated that prior to Analysis outliers were screened. In line 186 it has been stated that no outliers were identified. Can Authors comment in further detail of what criteria was used for screening the outliers.

2) In section 2.1, details about profession, educational level, gender and age have been described. It will be interesting to see if Authors observe any particular pattern in results based on profession or gender.

Reviewer 2 Report

The aim of the present study was to examine the relationship between self-compassion, work motivation and mental health problems and investigate which type of work motivation could mediate between self-compassion and mental health problems. This online survey study was conducted using a sample of Japanese workers (n = 165) with a cross-sectional design. Correlation and path analyses were performed for data analysis. Correlation results showed that self-compassion was significantly and negatively correlated with amotivation and mental health problems while amotivation was significantly and positively correlated with mental health problems. Path analysis results showed that amotivation rather than intrinsic or extrinsic motivation mediated the impact of self-compassion on mental health problems. Conclusion was drawn that self-compassionate people tended to be less amotivated and had fewer mental health problems. The present study contributes to better understanding how self-compassion may benefit mental health and suggests future self-compassion training can be targeted at amotivated workers.

The authors present an interesting research question concerning mental health problems at the workplace. However, please further consider the concerns listed below, to further improve this paper.

First, in the introduction section, the authors mentioned cultural differences in mental health and pointed out the meaning of conducting research in the non-WEIRD background. However, in the discussion section, the authors didn’t mention if there’re actual differences in different cultures. As a response to the foregoing cultural difference, it’s recommended to report mental health in Japan as reflected in the sample, and compare it with mental health as reported in other cultures.

Second, (a minor point) the scale used to measure mental health problems in this study consists of depression, anxiety and stress. Thus, when introducing the relationship between mental health problems and other variables, it would be better to mention the specific mental health problem rather than using “mental health” generally.

Third, it’s acceptable for the authors to introduce the main constructs one by one but it should be noted that evidence supporting the relationship between variables is insufficient. For example, the authors introduced the relationship between self-compassion and mental health problems simply in just a few words (“The 70 importance of self-compassion in relation to mental health has been reported in many 71 different cultures including Japan”). The relationship between these two variables should be further discussed. More references should be added to support the positive effect of self-compassion on mental health (e.g. interventional studies). Furthermore, since the aim of the current study was to find the potential motivation mediator between self-compassion and mental health, mediators that have been already tested by previous researchers should be introduced.

Fourth, my major concern is about the sample. Though the sample has its particularity targeting at Japanese workers, I have to say the sample size is relatively small (n = 165) in cross-section studies. Besides, the data used in the present study is not original as it has been used in another study. I know authors had conducted the study, but please consider to clarify this limitation. 

Fifth, I have a question about the reliability of the scale WEIMS. As the authors reported, the smallest Cronbach’s α of the subscale is 0.64. I wondered whether this value can demonstrate adequate reliability.

Round 2

Reviewer 1 Report

Authors have addressed all my comments and incorporated necessary changes. I recommend this for publication in present form.

Reviewer 2 Report

Thanks for your efforts in improving the article. I'm satisfied with the revision.